# In Vitro Inducted Tetraploid *Elsholtzia splendens* Nakai ex F. Maek. Alters Polyphenol Species and Synthesis

**DOI:** 10.3390/plants13233374

**Published:** 2024-11-30

**Authors:** Jie Liu, Dang Yang, Xin Li, Zexin Jin, Junmin Li

**Affiliations:** Zhejiang Provincial Key Laboratory of Plant Evolutionary Ecology and Conservation, School of Life Sciences, Taizhou University, Taizhou 318000, China; 18305769558@163.com (D.Y.); lixin2013@tzc.edu.cn (X.L.); jzx@tzc.edu.cn (Z.J.)

**Keywords:** *Elsholtzia splendens*, tetraploid, 8-zonocolpate pollen grains, polyphenols, tandem duplication, neofunctionalisation

## Abstract

*Elsholtzia splendens* Nakai ex F. Maek. has been employed in traditional Chinese medicine for millennia. Nevertheless, the small size and the paucity of research on its pharmacological effects have restricted its extensive utilisation in clinical medicine. Polyploid breeding represents an effective method for the rapid enhancement of plant biomass and metabolites. In this study, the most effective in vitro method for inducing tetraploid formation was identified as axillary buds treated in a solution of colchicine at a concentration of 1% for 24 h. Meanwhile, a comparison between tetraploids and diploids yielded two significant findings: (1) The presence of 6-zonocolpate and 8-zonocolpate pollen grains can be used as distinguishing characteristics for diploid and tetraploid, respectively. (2) Genome duplication resulted in alterations to the polyphenol species and synthesis pathway in *E. splendens*. The accumulation of wogonin, oroxylin A, baicalin, chrysin, acacetin and related derivatives was markedly greater in tetraploid plants, whereas apigenin, naringenin, scutellarein and related derivatives were found to accumulate to a greater extent in diploid plants. It is noteworthy that wogonin and oroxylin A were uniquely detected in tetraploids, indicating that the generated tetraploids may harbor novel pharmacological value. The findings not only provided new insights into the metabolic mechanism of polyploidisation but also established a foundation for the selection and breeding of novel genetic resources of *E. splendens*.

## 1. Introduction

*Elsholtzia splendens* Nakai ex F. Maek (*E. splendens*) is an annual herb belonging to the *Elsholtzia* genus of the Labiatae family. It is primarily distributed along the middle and lower reaches of the Yangtze River in China [1] and Korea [2]. Most researchers are currently focusing on its remarkable ability to absorb and accumulate the heavy metal copper in soils [3,4], which has led to its status as a highly sought-after Cu hyperaccumulating plant [5,6]. Indeed, it was initially employed in traditional Chinese medicine as an antiviral, antibacterial, antispasmodic, and asthma treatment, and is frequently utilised as an ingredient in folk medicines for invigorating blood circulation and removing blood stasis [7], which can be traced back millennia [8]. Nevertheless, its pharmacological value has not been extensively employed in clinical practice. The absence of pharmacological action studies may be a contributing factor to the limited widespread use of this species. Additionally, its small size may present a further challenge.

How can we rapidly enhance the biomass and effective components of *E. splendens*? Polyploidisation may be proved to be the optimal solution. A substantial body of research has demonstrated that the expression of repetitive genes underwent alterations in polyploids as a consequence of the gene dose effect [9,10,11]. Consequently, polyploid plants exhibited notable differences from their diploid progenitors in terms of growth and development, morphology and physiology, photosynthetic efficiency, genetic adaptation, metabolites and adaptability to environmental stress [12,13,14,15,16,17,18,19]. Among them, there is no shortage of examples where polyploidisation has increased the amount of active metabolites. For example, the tetraploid *Anoectochilus formosanus* Hayata exhibited significantly elevated levels of total flavonoids and gastrodin in the entire plant relative to the diploid [18]. It was documented that polyploidisation enhanced the production of cannabidiol and terpenes in the *Cannabis sativa* L. plant [19]. The phenolic content of the tetraploid *Cnidium officinale* Makino was found to be higher than that of the diploid [20]. Additionally, the novel phenotypes and functions possessed by many polyploid plants frequently exhibited greater practical applications in production practices, as evidenced by the tetraploid wallflower (*Erysimum cheiri* (L.) Crantz) [9], the tetraploid *Morus alba* L. [21], the tetraploid citrus (*Citrus limonia*) [22] and so on. It is evident that the polyploidisation of *E. splendens* represented a viable method for increasing biomass and effective components. The newly created polyploid germplasm has the potential to rapidly increase the content of active ingredients and may even facilitate the formation of new compounds.

Mitotic inhibitors, including colchicine, oryzoline, triflumizole, and amiprophos-methyl, have been extensively employed as chromosome-doubling agents [23]. Colchicine is the most commonly employed agent for the induction of polyploidy, due to its ability to inhibit mitosis, destroy the spindle, and halt the progression of chromosomes at the mid-division stage, thereby facilitating chromosome doubling. A number of plants have been treated with colchicine in order to obtain new polyploid material. Nevertheless, there are no documented instances of the *Elsholtzia* genus other than *E. splendens*. Only tetraploid *E. splendens* was previously obtained through the primary meristem processing technique in our laboratory [24]. Furthermore, the induction of polyploids using the tissue culture method has not been documented in any *Elsholtzia* genus. Accordingly, the present study employed wild diploid *E. splendens* from China for the purposes of colchicine tissue culture and the development of an identification technology system. The objective was to offer the most optimal solution for in vitro polyploidisation of *Elsholtzia* and other closely related species from multiple perspectives, thereby facilitating the accurate and rapid identification of new polyploid germplasm. More importantly, the induced polyploids can be utilised to investigate changes in the composition and concentration of active compounds in the *E. splendens* polyploids, with a view to identifying additional valuable metabolites and enhancing their pharmacological value for practical application in clinical and field production.

## 2. Results

### 2.1. Screening of Axillary Bud Regeneration and Rooting Medium

Nodes containing young axillary buds were extracted from healthy mother plants. Following approximately 10 days of pre-culture, the specimens were transferred to an MS differentiation medium containing varying concentrations of growth regulators (Table 1). The formation of healing tissues was observed in the vicinity of the incision site. The number of incisions and the number of healing tissues were quantified in order to determine the extent of indeterminate branch regeneration after a period of 50 days. A series of concentration ratios were established to identify the optimal induction conditions for axillary bud explants of *E. splendens*. As illustrated in Table 1, this phenomenon may be attributed to the heightened sensitivity of the plant, whereby elevated levels of N-(Phenylmethyl)-9H-purin-6-amine (6-BA, above 0.1%) impeded the differentiation of adventitious shoots. The ratio of the concentration of cytokinin to that of the growth hormone also had an effect on the regeneration of adventitious shoots. The findings indicated that a 6-BA to 1-Naphthylacetic acid (NAA) concentration ratio between 5:1 and 10:1 was more efficacious in promoting plant regeneration. The results demonstrated that the optimal conditions for adventitious shoot induction were MS + 6-BA 0.3 mg/L + NAA 0.03 mg/L + agar 6.5 g/L + sucrose 20 g/L, which resulted in a regeneration rate of 95.72%. Similarly, different hormone types were employed for the induction of rooting culture, and the maximum number of roots and the number of days required to reach this maximum were quantified (Table 2). The most rapid rooting medium was identified as MS + Indole-3-butyric acid (IBA) 0.1 mg/L + NAA 0.1 mg/L + agar 6.5 g/L + sucrose 20 g/L. Although the aforementioned rooting medium facilitates the rapid production of roots in substantial quantities, the resulting roots are brittle and susceptible to fracture during transplantation, which has a negative impact on the survival rate. Accordingly, 1/2 MS medium devoid of any hormones was ultimately elected to pursue a solution.

### 2.2. Screening of Tetraploid Induction Medium

As illustrated in Table 3, following the application of three concentration gradients (0.02%, 0.05%, 0.1%, and 0.2%) of colchicine and three time intervals (6 h, 12 h, and a total of 24 h) of orthogonal experiments were conducted, and the optimal condition for the induction of tetraploid in *E. splendens* was identified as the treatment with 0.1% colchicine for 24 h, resulting in an induction rate of up to 88.56%. Although treatment with 0.05% or 0.02% colchicine resulted in a higher survival rate, the number of chimeras was markedly elevated. It is possible that the plant is highly sensitive to colchicine, with a significant increase in mortality at a concentration of 0.2%, and the survival rate of successfully induced explants was low.

### 2.3. Comparison of Phenotypic Differences Between Diploids and Tetraploids

At the tissue culture stage, significant differences in plant and leaf shape were observed between diploids and tetraploids (Figure 1a). In comparison to the diploids, the tetraploid plants exhibited a greater degree of serration at the margins of their leaves, with a thicker and uneven leaf blade, a reduced leaf aspect ratio, and an overall larger plant size (Figure 1b). Upon transplantation to soil or hydroponics, the distinctions in morphology became more pronounced (Figure 1c). In addition to the variations in leaf blade morphology (Figure 1d), the root system exhibited a greater number of thick lateral roots (Figure 1e), while the inflorescences displayed increased length (Figure 1f) and density of the per-floret hairs (Figure 1g). Significant differences were observed in 14 morphological characters between the two ploidy levels of *E. splendens* in adult plants (Table 4).

### 2.4. Flow Cytometry (FCM) and Karyotyping Identification of Diploids and Tetraploids

The FCM analysis demonstrated a peak fluorescence intensity value of approximately 2.25 × 10^5^ for the tetraploid cells, which was twofold greater than that observed for the wild-type diploid (Figure 2a,c). The number of chromosomes in the diploid samples was determined through 4′,6-diamidino-2-phenylindole (DAPI) fluorescence staining, which revealed 16 chromosomes with a length of 1.0–2.0 microns (Figure 2b). The chromosomes are primarily located in the near-central centromeric region and have a small genome. The same method was employed to obtain 32 chromosomes in tetraploids (Figure 2d). The chromosome specimen of the diploid samples was subjected to fluorescent in situ hybridization (FISH) using the telomere repeat probe, which further defined the number of chromosomes in the material as 16 (Figure 2e). Additionally, the 5SrDNA and 18SrDNA repeat probes (Figure 2f) demonstrated that all samples exhibited six chromosomes with a prominent 5S rDNA (red) hybridization signal and six chromosomes with a strong 18S rDNA (green). The results thus corroborate the hypothesis that the detected samples are diploid, exhibiting a karyotype of 2n = 2x = 16. The chromosome specimens of the tetraploid samples were subjected to the same methodology. The telomere repeat probe (Figure 2g) and 5SrDNA hybridization signal (Figure 2h), respectively, indicated that the samples exhibited a 2n = 4x = 32 karyotype and were polyploid.

### 2.5. Microstructural and Ultrastructural Comparison Between Diploids and Tetraploids

The identification of plant ploidy through the measurement of stomatal guard cell length was demonstrated to be a straightforward and viable approach for the assessment of chromosome ploidy in regenerating plants. Given that, scanning electron microscopy (SEM) of the stomata was conducted on transplants that were verified by the karyotyping procedure. As illustrated in Figure 3, the stomatal size of tetraploid leaves was markedly larger than that of diploid leaves, and the guard cells were also more pronounced (Figure 3a,b). In the course of the actual measurements, the stomatal length and width of tetraploid leaves were significantly larger than diploids (Figure 3c). Interestingly, it was noted that the stomatal density of tetraploid leaves increased with increasing ploidy (Figure 3c). The number of stomata per square millimetre of leaf was 160–190 for diploids, and 200–225 for tetraploids (Figure 3c). This phenomenon was the antithesis of the norm, as the typical pattern observed was that of a reduction in stomatal density with increasing ploidy.

The diploid parents of *E. splendens* were primarily 6-zonocolpate pollen grains, characterised by clear, elongated and deep pollen colpus that extend to the poles of the grain and exhibit polar axis symmetry (Figure 3d,f). The polyploid lines predominantly exhibited 8-zonocolpate pollen grains, characterised by wider and shallower colpus, displaying left–right symmetry (Figure 3e,g). There were notable disparities in the morphology of the pollen furrows between tetraploidy lines and their diploid progenitors. Pollen morphology can also be employed as a significant indicator for the assessment of chromosome ploidy in *E. splendens*.

Transmission electron microscopy (TEM) was used to observe the chloroplasts ultrastructure of two different ploidy-level *E. splendens*. The findings indicated that the number of chloroplasts within each cell, the dimensions of the chloroplasts, and the quantity and size of the starch granules within each chloroplast increased with rising ploidy levels. Furthermore, the thylakoids and granule lamellas were observed to be more closely and neatly arranged in the tetraploid chloroplasts in comparison to the diploids (Figure 3h,i).

### 2.6. Metabolite Comparison Between Diploids and Tetraploids

Ultra performance liquid chromatography/tandem mass spectrometry (UPLC-MS/MS), as a broadly targeted metabolomics approach, was used to detect and quantify the metabolome between the two ploidy levels of *E. splendens* in this study (Figure 4). Overlapping peaks in the total ion current were observed in different quality control (QC) samples (Figure 4a,b). The results showed overlapping and consistent peaks in ion intensity (shown on the *y*-axis) and retention times (shown on the *x*-axis) and the stability of the peaks for the same samples at different time intervals.

The structure of the data was evaluated using principal component analysis (PCA). The PCA plots demonstrated the existence of two distinct groups, based on the distribution of metabolite ion intensities in specific tissues (Figure 4c). The grouping of biological replicates in each tissue was found to be tight, indicating that high-quality metabolomic data were obtained.

A total of 1073 metabolites were identified through analysis and subsequently classified into 12 principal classes and 44 subclasses (Appendix A). In order to gain insight into the metabolic properties of two different ploidy-level *E. splendens*, a differentially accumulated metabolites (DAMs) analysis was performed. This revealed that 240 DAMs exhibited distinct accumulation patterns between the leaves of the two materials (Appendix A). Figure 4d illustrates the differential accumulation tendency of these identified DAMs. In comparison to the dipolids, of the 144 DAMs that were found to accumulate in an up-regulation in the tetraploids, 96 DAMs were found to be down-accumulated (Figure 4d and Appendix A). The identified DAMs can be classified into 11 major metabolite categories, namely flavonoids, phenolic acids, other metabolites, organic acids, lipids, terpenoids, lignans and coumarins, amino acids and their derivatives, nucleotides and their derivatives, and alkaloids and tannins. Of these, flavonoids and phenolic acids were the most abundant, collectively representing 50.21% of all differential metabolites (Figure 4e).

Following an in-depth analysis, it was observed that, in comparison to the diploids, a number of substances, including wogonin, 3′,7-dihydroxy-4′-methoxyflavone, 3,3′,5-trihydroxy-4′,7-dimethoxyflavanone, oroxylin A, betulin, chrysin, quillaic acid, coniferin, methyl anthranilate and baicalin, demonstrated a greater accumulation in tetraploids (Figure 4f and Appendix A). It is worthy of note that the top seven substances were not detected in diploids. Similarly, it was observed that certain substances, including hesperetin-5-O-glucoside, eriocitrin, cyclo (Pro-Pro), and 3-hydroxymandelate, were undetected in tetraploid samples but exhibited elevated accumulation in diploids (Figure 4f and Appendix A). This outcome may indicate that alterations in ploidy have initiated modifications in specific metabolic pathways. In particular, the accumulation of methoxyflavonoids in tetraploids was observed to be the most significant among all substances. The aforementioned substances exhibited a comparable carbon skeleton structure (Figure 4g), which may indicate a similar structure–activity relationship. Additionally, the accumulation of terpenoids (betulin, quillaic acid, asiatic acid), LysoPE (18:1, 17:1, 16:0, etc.), and free fatty acids ((7Z)-hexadecenoic acid, vaccenic acid, petroselinic acid, linoleic acid) was also observed in tetraploids to varying degrees above diploids (Appendix A).

## 3. Discussion

*E. splendens* is widely utilised for investigation in a variety of studies on soil remediation, due to its distinctive capacity to serve as a copper indicator. Nevertheless, due to its limited biomass and narrow range of suitability, its potential applications in fields such as medicine, food, spices, insecticides, detergents, animal feed, and others remain to be fully explored. In this study, a polyploid tissue culture system of *E. splendens* was constructed for the first time. This approach allows the induction of polyploids not only via direct seedling but also through in vitro regeneration, thereby meeting the diverse requirements of the market. It was determined that maintaining a balance of cytokinin and auxin ratio was of great importance during the in vitro regeneration process, a finding that was consistent with the results of numerous previous studies. It is recommended that the ratio of cytokinin to auxin be maintained between 10:1 and 5:1 to ensure optimal growth of regenerating buds. The same result was corroborated by the study of Zhang et al. [25]. The process of polyploid mutagenesis necessitates the precise regulation of two pivotal variables: the concentration of the mutagen and the duration of the treatment. The latter is contingent upon the tolerance level of the plant, whereas the former is inextricably linked to the cell division cycle of the plant. The concentration of *E. splendens* employed in the treatment was low and the duration of exposure brief due to the plant’s sensitivity to mutagens and the minute size of its chromosomes. The addition of hormones facilitates faster growth during the process of rooting culture; however, this resulted in the roots becoming brittle and significantly reduced the survival rate of transplantation. Conversely, the omission of hormones and the utilisation of a 1/2 MS medium to reduce the inorganic salt concentration could facilitate the development of flexible roots and enhance the survival rate of transplantation.

Flow cytometry and chromosome number determination are two widely employed techniques for the identification of ploidy levels in plants [13,23,25]. However, the small chromosomes of *E. splendens* can facilitate the overlap of chromosomes, which can result in an erroneous count of chromosomes, particularly when the ploidy level is elevated. To address this issue, we employed the FISH technique in order to ascertain the number of polyploid chromosomes in *E. splendens* with greater accuracy [26,27]. The implementation of FISH resulted in a notable improvement in the resolution of chromosomal karyotype analysis, thereby providing a more accurate representation of the chromosomal makeup of the species. Additionally, our study revealed that alterations in pollen morphology may serve as a distinguishing characteristic of tetraploids. This feature was similarly presented in other species.

The effects of polyploidy on plant species are frequently unpredictable and variable. Nonetheless, a considerable corpus of research has substantiated the proposition that an increase in chromosome number has the effect of enhancing a plant’s overall nutritional status and metabolic potential [25,28,29,30]. The present study demonstrated that tetraploids produced a greater quantity of secondary metabolites than wild diploids, with the majority of these changes occurring in the flavonoids and phenolic acids, including wogonin (5,7-Dihydroxy-8-methoxyflavone), 3′,7-dihydroxy-4′-methoxyflavone, 3,3′,5-trihydroxy-4′,7-dimethoxyflavanone, oroxylin A (5,7-dihydroxy-6-methoxyflavone), chrysin (5,7-Dihydroxyflavone), baicalin (7-D-Glucuronic acid-5,6-dihydroxyflavone), 3,7-dihydroxy-4′-methoxyflavone, 5,6,7-tetrahydroxy-8-methoxyflavone, pinocembrin (5,7-Dihydroxyflavanone), acacetin (5,7-dihydroxy-4′-methoxyflavone), chrysoeriol (4′,5,7-Trihydroxy-3′-methoxyflavone), izalpinin (3,5-Dihydroxy-7-methoxyflavone), and so on. Among them, the first four substances were newly discovered in tetraploids and were not detected in diploids, which all belonged to methoxylated flavones. As documented in the literature, methoxyflavonoids were biosynthesized by the secretory tissues of plants belonging to the Lamiaceae, Asteraceae, and Cruciferae families and stored externally or in oil cavities [31]. These compounds were demonstrated to exhibit a range of in vitro and in vivo biological activities [31,32]. They not only played roles in phytochemical defence mechanisms but also served as a natural source of lead molecules for the development of potent antiproliferative, antidiabetic or anti-inflammatory drugs [31]. It is worthy of note that a substantial body of evidence from in vitro and in vivo experiments has demonstrated that methoxyflavones can inhibit the growth of various cancer cells and display selective toxicity towards malignant cells, which have the potential to play a significant role in the field of cancer therapy and chemoprevention [31,33]. Wogonin was demonstrated to possess a range of biological activities, including anti-tumour, central nervous system, anti-inflammatory, antiviral, antimicrobial and hypoglycaemic properties [34]. Of particular note is its impact on anti-tumour therapy, which has been significant with no serious adverse effects [35]. The structure of 3′,7-dihydroxy-4′-methoxyflavone (farnisin) exhibited a high degree of similarity to that of acacetin, which was initially identified in the seeds of *Acacia farnesiana* [36]. Despite the dearth of literature on this topic, we postulated that its function was analogous to that of acacetin, which was demonstrated to induce cancer cell cycle arrest and apoptosis and autophagy [37]. The compound 3,3′,5-trihydroxy-4′,7-dimethoxyflavanone was reported to exhibit enhanced activity against both *S. aureus* and *S. typhi* [38]. Oroxylin A was demonstrated to possess a range of beneficial properties, including anti-inflammatory, anti-tumour, blood vessel and nerve cell protective, memory enhancing, antiviral, and other effects [39]. In particular, it was demonstrated that this compound significantly enhanced cognitive and mental function in animal models of brain ageing and neurodegeneration [33]. It is, therefore, anticipated that the production of these substances may result in the modification of the medicinal–pharmaceutical properties of tetraploid *E. splendens* and is expected to improve its application in the field of cancer therapy.

What is more, the findings of this study indicated that wogonin, oroxylin A, baicalin, chrysin, and acacetin were markedly accumulated in tetraploids, whereas a greater accumulation of apigenin, luteolin, naringenin, and scutellarein was observed in diploids. Despite the similarity in their carbon backbones, the accumulation of these metabolites varies significantly between plants of different ploidy levels. It can be observed that the ploidy level did not exert a uniform influence on the accumulation of these substances, which were composed of similar elements. This prompted further interest from the research team, we sought to understand how these substances accumulated and what the synthetic pathway might be. Following an exhaustive review of the existing literature, two distinct pathways for flavonoid synthesis in *Scutellaria baicalensis* were identified as meriting further investigation [32,40,41,42]. The synthesis of flavonoid in the aboveground parts of the plant occurred via a classical pathway involving a number of enzymes, including phenylalanine deaminase (SbPAL), cinnamate 4-hydroxylase (SbC4H), 4-coumarate coenzyme A (CoA) ligase (SbCLL1), chalcone synthase (SbCHS-1), chalcone isomerase (SbCHI) and flavonoid synthase II (SbFNSII-1) [41,42]. This resulted in the production of the 4′-hydroxyflavone apigenin, which subsequently underwent hydroxylation and glycosylation to form scutellarein and scutellarin, respectively [41,42]. In contrast, an additional specialised pathway was observed in the root parts of the *Scutellaria baicalensis* plant [40,41,42]. The initial distinction between these two pathways was based on the function of cinnamate-CoA ligase. SbCLL-7 (a cinnamate-CoA ligase) was directly recruited from fatty acid/jasmonate biosynthesis to form cinnamoyl-CoA from cinnamate in the root-specific pathway [43]. Subsequently, SbCHS-2 (SbCHS-1 isoformer), SbCHI and SbFNSII-2 (SbFNSII-1 isoformer) cooperated to form chrysin, a member of the 4′-deoxyflavone family of compounds [32]. Finally, modifications were made by flavonoid hydroxylases, methyltransferases, glycosyltransferases, and O-methyltransferases, resulting in the formation of wogonin, baicalin, and other derivative compounds [40]. It was noteworthy that the polyphenols produced by the two pathways exhibited similarities to those accumulated in the two ploidy levels of *E. splendens*, respectively. The polyphenols produced by the aboveground parts pathway exhibited similarities to those of the diploids, whereas the metabolite productions of the root parts were similar to those of the tetraploids. This prompted the formulation of a hypothesis: could the ploidy level be influencing a photosynthetic pathway, thereby affecting polyphenol synthesis?

A substantial body of research has demonstrated that light exerts a regulatory effect on key enzymes involved in polyphenol metabolism. The activity of enzymes involved in polyphenol metabolism was influenced by the quality and intensity of light, which in turn affected polyphenol synthesis [44]. Furthermore, our previous research proved that polyploidy had a significant effect on the photosynthetic capacity of *E. splendens*. It would be meaningful to ascertain whether there is an inherent correlation between these two phenomena. However, previous reports have indicated that baicalin and baicalein are not exclusively produced in the roots of *S. baicalensis* but are also found in leaves [45]. It was, therefore, evident that the alteration in synthase nature in the polyphenol roots of *S. baicalensis* was not merely a consequence of the presence or absence of light. In 2019, Zhao et al. demonstrated that the 4′-deoxyflavonoid pathway evolved through genomic changes including gene mutations (SbCLL-7), gene duplications and neofunctionalisation (SbCHS-2, SbFNSII-2, SbF8H and SbPFOMT), gene amplification (SbCHS-2), segmental duplications (SbFNSII-1), tandem duplications and neofunctionalisation (CHS-2, FNSII-2) and tandem duplication and subfunctionalisation (SbPFOMT5) in genus *Scutellaria* [32]. Thus, the analogous phenomenon observed in diploid and tetraploid *E. splendens* was not merely attributable to alterations in photosynthetic capacity resulting from ploidy, which then influenced the synthesis of polyphenol species. It was possible that the doubling of the genome may have resulted in tandem duplication or neofunctionalization in certain genes that regulate polyphenol synthesis in tetraploid *E. splendens*. This may be the most plausible explanation for the alteration of polyphenol types in tetraploid species. Additionally, methyl jasmonate was observed to be markedly accumulated in the tetraploid relative to the diploid. Schneider et al. [46] reported that SbCLL-7, which specifically ligates CoA to cinnamate, evolved from a CoA ligase involved in fatty acid/jasmonate biosynthesis. It could thus be surmised that the alteration in the ploidy level of *E. splendens* might result in modifications to the activity of transcription factors that influence the jasmonic acid synthesis pathway, which subsequently affected the polyphenol synthesis pathway. This is a complex question, and much more experimentation was required to elucidate the underlying mechanisms. A schematic diagram of the mechanisms involved was presented in order to illustrate the reasons for the ploidy-influenced polyphenol species observed in *E. splendens* (Figure 5).

## 4. Materials and Methods

### 4.1. Materials Cultivation and Colchicine Treatment

#### 4.1.1. Material Cultivation

The seeds of the wild *E. splendens* were gathered from Hong-an County (31°30.632′ N, 114°32.620′ E), Hubei province, China, in 2013. The fourth generation of seeds from successive plantings were employed in this experiment. The seeds were planted in a soil mixture comprising peat, vermiculite, pine bark and sand in a ratio of 6:2:2:1, respectively [24]. The stem tips of the plants were harvested and employed as materials for tissue culture with colchicine treatment when three pairs of true leaves had emerged.

#### 4.1.2. Establishment of the Regeneration System of Diploid *E. splendens*

The methodology refers to the description of Zhang et al. [25]. Containing young axillary buds were extracted from healthy mother plants, sterilised in a 1% sodium hypochlorite and 0.5% (*v*/*v*) Tween-20 solution for a period of five minutes, and then rinsed three times in sterile distilled water for a duration of one minute each time. In view of the sensitivity of *E. splendens* plants, an orthogonal assay was conducted using sixteen combinations of 6-BA (0.1 mg/L, 0.3 mg/L, 0.6 mg/L, 1 mg/L) and NAA (0.01 mg/L, 0.03 mg/L, 0.06 mg/L, 0.1 mg/L) to investigate the effects of different hormone types on the induction of young stem healing tissue. The pH of all medium was adjusted to 6 with 1 M KOH or HCl and then autoclaved at 120 °C for 20 min. Similarly, different hormone combinations, 1/2 MS or MS + IBA (0 mg/L, 0.05 mg/L, 0.1 mg/L) + NAA (0 mg/L, 0.05 mg/L, 0.1 mg/L), were used for the induction of rooting culture, and the rooting percentage and root length were counted after 30 days. The optimal rooting medium was identified after 30 days. The same media screened above were used for polyploid culture conditions.

#### 4.1.3. In Vitro Culture with Colchicine Induction

In this experiment, colchicine was selected as the mutagen, and the immersion method was employed to induce the treatment of wild *E. splendens* stem tip segments. Four concentration gradients (0.02%, 0.05%, 0.1%, and 0.2%) of colchicine and four time intervals (6 h, 12 h, 24 h and 48 h) of orthogonal experiments were used, respectively. The treatment medium was prepared by diluting the stock solution (1 g of colchicine dissolved in 75% ethanol), which was obtained by separately diluting the stock solution into 20 mL of liquid MS medium (containing 20 g/L sucrose, pH 6). To prevent photodegradation of colchicine, the cultures were covered with tin foil. During the mutagenic treatment, the cultures were incubated in the dark at 25 °C on a shaker at 90 rpm. Following the colchicine treatment, the materials were washed three to five times with sterile water and inoculated on the induction medium, which was identified as the optimal induction medium for diploids. Five axillary buds were placed obliquely on the medium and cultured in the dark (room temperature 25 °C) for regular observation and statistical analysis.

### 4.2. Chromosome Ploidy Level Analysis

#### 4.2.1. Flow Cytometry (FCM) Measurement

FCM adopted the method description of Liu et al. [24]. Approximately 0.2 g of fresh leaves were placed in 1 mL of pre-cooled Galbraith’s buffer (0.1% (*V*/*V*) Triton-100, 20 mmol/L MOPS, 45 mmol/L MgCl_2_, 30 mmol/L sodium citrate, 0.5% (*V*/*V*) Polyvinylpyrrolidinone K-30, pH 7.0), then rapidly chopped into a paste with a sharp blade and filtered through a 30 μm nylon filter mesh. Subsequently, the filtered homogenate was treated with RNAase for 10 min. Next, the cells are stained with 50 μL/mL propidium iodide (PI) for 30 min. Following staining, the cells were analysed using a flow cytometer (Attune™ Acoustic Focusing Cytometer, Applied Biosystems ABI, Foster City, CA, USA), which measures the fluorescence intensity that correlates with DNA content. The data were displayed as a DNA histogram, with peaks corresponding to different ploidy levels. By analysing these peaks, the extent of polyploidy in the sample can be quantified.

#### 4.2.2. Karyotype Analysis

After FCM screening, about 1.5 cm root tips of screened materials were excised when the plant roots reached a length of 2–3 cm, in order to facilitate karyotyping identification. The specific operations referenced description in Zhang et al. [27]. Initially, the root tips were placed in N_2_O at 0.9–1.0 MPa for 2 h to obtain a substantial number of cells in the mitotic phase. Next, the roots were fixed with 90% pre-cooled glacial acetic acid for 10 min, whereafter the white part of the root tip was excised with a razor blade and enzymatically digested in a 3:1 mixture of cellulase and pectinase at 37 °C for 70 min. Subsequently, the enzyme-digested root tips were fully fragmented with a dissecting needle in order to create a cell suspension with glacial acetic acid. Finally, chromosome-dispersed cells in the meiotic phase were counterstained and enumerated using 4′,6-diamidino-2-phenylindole (DAPI). What is more, fluorescence in situ hybridisation was also conducted using the telomere-specific repeat probe Oligo-(TTTAGGG)_6_ and the 5S rDNA/18S rDNA fluorescent probe, respectively, to further clarify the chromosome number and ploidy characteristics. The resulting images were captured using a fluorescence microscope (Ci-s Nikon, Tokyo, Japan).

### 4.3. Phenotypic and Microstructural Ploidy Level Analysis

#### 4.3.1. Morphological Traits Evaluation of Different Organs

All *E. splendens* plants generated were sampled for the purpose of characterizing the variation between diploid and tetraploid plants. A series of morphological characteristics were employed to differentiate between the plants, including plant height, number of branches, dry weight, fresh weight, root length, shoot length, shoot diameter, leaf length, leaf width, leaf length/leaf width ratio, specific leaf area (SLA), specific leaf weight (SLW), flower length and width. The aforementioned morphological characteristics were recorded with the aid of a camera. The area of individual leaves was measured using the software Image J (https://imagej.net/ij/download.html accessed on 15 September 2024). Leaf area and dry mass were used to calculate the following parameters:

Specific Leaf Area (SLA) = leaf fresh area/leaf dry weight (cm^2^/g), was calculated according to Cornelissen et al. [47].

Specific Leaf Weight (SLW) = leaf dry weight/leaf fresh area (g/cm^2^), was calculated according to Cornelissen et al. [47]. Specific leaf weight is the inverse of the specific leaf area.

#### 4.3.2. Microstructural and Ultrastructural Evaluation of Leaf and Pollen

The differences in leaf trichomes, leaf stomata, and pollen between two ploidy level materials were discerned through the utilisation of scanning electron microscopy (SEM). The method was modified based on description of Zhang et al. [48]. About 1 mm^2^ leaf blades or 3–4 flower spikelets were fixed in a FAA solution at 4 °C for 24 h. On the following day, the fixed material was dehydrated in a stepwise gradient of alcohol (30%, 50%, 70%, 80%, 90%, and 95% alcohol in water) to pure alcohol, followed by critical point drying (EM CPD300, Wetzlar, Leica, Germany). Subsequently, the samples were mounted on aluminium columns coated with a double layer of conductive adhesive and were sprayed with gold–palladium. Finally, the state of the leaf trichome stomatal and pollen were observed with a scanning electron microscope (Phenom Pro, Phenom-world, Eindhoven, The Netherlands).

Chloroplast ultrastructure observation was performed using transmission electron microscopy (TEM) outlined by Deng et al. [49] with certain modifications. Approximately 1 mm^2^ Leaves were fixed in 2.5% glutaraldehyde solution (9 mL 1 × PBS phosphate buffer + 1 mL 25% glutaraldehyde, pH 7.4) at 4 °C for 6 h. After three times rinse in phosphate-buffered saline (PBS) for 30 min each, fixed samples were immersed in osmotic acid staining solution (1% osmotic acid + 0.1 mol/L PBS) overnight. Subsequently, each sample was rinsed again four times in PBS phosphate buffer, and then dehydrated in an ascending alcohol series. The samples were then immersed in a series of propylene oxide and embedding agents, and polymerised at 65 °C for 48 h. Finally, ultrathin sections were obtained using a Leica UC7 ultramicrotome and stained with uranyl acetate and lead citrate. The ultrastructure of the chloroplasts was observed using a transmission electron microscope (Tecnai G2 F20 S-TWIN, FEI, Hillsboro, OR, USA).

### 4.4. Metabolite Variance Evaluation

#### 4.4.1. Sample Preparation and Extraction

As Zhang et al. [25] described, the biological samples were subjected to lyophilisation using a vacuum lyophiliser (Scientz-100F, Ningbo, China) firstly. Then, the lyophilised samples were pulverised using a zirconia bead mixer (MM 400, Retsch, Haan, Germany) at 30 Hz for 1.5 min. One hundred milligrams of the lyophilised powder was dissolved in 1.2 mL of a 70% methanol solution, and vortexed for 30 s every 30 min for a total of six times. The samples were then stored overnight in a refrigerator at 4 °C. Subsequently, the extract was subjected to centrifugation at 12,000 rpm for 10 min, after which it was filtered through 0.22 μm organic phase nylon needle filter (SCAA-104, ANPEL, Shanghai, China) before UPLC-MS/MS analysis.

#### 4.4.2. UPLC Conditions

Sample extracts were analyzed using a UPLC-ESI-MS/MS system (UPLC: Shimadzu Nexera X2; MS: Applied Biosystems 4500 Q TRAP, MA, USA). The analytical conditions consulted description of Xiao et al. [50]: The UPLC system employed a column of Agilent SB-C18 (1.8 µm, 2.1 mm × 100 mm) and utilised a mobile phase comprising solvent A (pure water plus 0.1% formic acid) and solvent B (acetonitrile plus 0.1% formic acid). The sample was determined using a gradient procedure, commencing at 95% A and 5% B. A gradient programmed in a linear fashion to 5% A and 95% B within nine minutes, and was maintained at this composition for one minute. Subsequently, the composition was adjusted to 95% A, 5.0% B in 1.1 min and maintained for 2.9 min. The flow rate was set at 0.35 mL per minute, the column temperature was set at 40 °C, and the injection volume was 4 μL. The effluent was alternatively connected to an electrospray ionisation (ESI) triple quadrupole linear ion trap mass spectrometer (QTRAP-MS).

#### 4.4.3. ESI-Q TRAP-MS/MS

Triple quadrupole-linear ion trap mass spectrometer (QTRAP™ AB4500 Q TRAP UPLC-MS/MS System, Waltham, MA, USA) was adopted to obtain linear ion trap (LIT) and triple quadrupole (QQQ) scans. The equipment contained an ESI Turbo Ion-Spray interface, which was operated in a positive and negative ion mode and the data were analyzed using analyst 1.6.3 software (AB SCIEX). During scanning, ESI source operation parameters were set as described by Tan et al. [51]: turbo spray (ion-source); 550 °C (source temperature); 5500 V (ion spray voltage or IS); ion source gas I (GSI), gas II (GSII), curtain gas (CUR) were set at 55, 60, and 25.0 psi, respectively; CAD (the collision gas) was high. The instrument was tuned and calibrated for mass, using 10 and 100 μmol/L polypropylene glycol solutions in QQQ and LIT modes, respectively. QQQ scans were conducted as multiple reaction monitoring (MRM) experiments, with a nitrogen collision gas set to a medium level. The DP and CE for each MRM transition were conducted with further DP and CE optimisation. A particular combination of MRM transitions was monitored for the duration of each period, according to the metabolites eluted within that period.

#### 4.4.4. Quality Control and Data Analysis

The mass spectrometry data were processed using the software Analyst 1.6.3. Principal component analysis (PCA) was conducted using the prcomp function within R (v4.3.1, http://www.r-project.org (accessed on 11 November 2023)), and the results of the hierarchical cluster analysis (HCA) of the samples and metabolites were presented as heatmaps with dendrograms. Pearson correlation coefficients (PCC) between samples were calculated using the cor function in R and presented as only heatmaps. The metabolites exhibiting significant regulation between groups were identified through the application of a VIP (variable importance in projection) value of ≥1 and an absolute log2FC (fold change) value of ≥1. The VIP values were extracted from the orthogonal partial lLeast squares discriminant analysis (OPLS-DA) result, which also encompassed score plots and permutation plots, this result was generated using the R package MetaboAnalystR. The identified metabolites were subsequently annotated using the KEGG Compound database (http://www.kegg.jp/kegg/compound/ (accessed on 11 July 2024)).

### 4.5. Statistical Analysis and Plotting

Two-factor analysis of variance (ANOVA) analysis and figure plotting were conducted by GraphPad Prism 8.0 software (GraphPad Software, La Jolla, CA, USA). *p*-values were corrected for multiple testing using the Benjamin–Hochberg (BH) method (*q*-value ≤ 0.05 was considered significantly enriched).

## 5. Conclusions

This study successfully established the *E. splendens* explant regeneration system and tetraploid induction system. The most effective method for inducing tetraploid formation was found to be soaking the axillary buds in a solution of colchicine at a concentration of 1% for a period of 24 h, and the survival rate of transplantation was also high. Subsequently, comparisons between tetraploids and diploids demonstrated that not only could stomatal size be employed as a discriminator between different ploidy-level *E. splendens*, but also 6- and 8-zoned pollen grains could also be used as one of the discriminators. At the same time, the type and content of flavonoids and phenolic acids present in the metabolites exhibited notable differences between the tetraploid and diploid samples. The accumulation of wogonin, oroxylin A, baicalin, chrysin, acacetin and related derivatives was evident in tetraploids, whereas in diploids, the accumulation of apigenin, naringenin, scutellarein and related derivatives was more pronounced. In particular, wogonin and oroxylin A were uniquely observed in tetraploids, indicating that the generated tetraploids may possess novel pharmacological properties distinct from diploids. Following an in-depth analysis, it was hypothesised that the alteration of polyphenol species between the diploid and tetraploid *E. splendens* may be attributed to the tandem duplication or neofunctionalisation of specific genes regulating different polyphenol synthesis, as a consequence of genome doubling. A schematic diagram of the mechanisms involved was constructed to illustrate the underlying reasons for the polyphenol species observed in *E. splendens* that were affected by the doubling process. The findings of this study provide new insights into the metabolic mechanism of polyploidisation, thereby establishing a theoretical basis for subsequent research in this field. Of greater significance, however, was that the generated tetraploids represent a valuable resource for investigating gene evolution within the same family and genus, as well as a foundation for the selection and breeding of novel genetic resources of *E. splendens*. Consequently, this provides a robust theoretical foundation for the integration of polyploidisation into the contemporary cultivation of pharmacologically valuable plants.

## Figures and Tables

**Figure 1 plants-13-03374-f001:**
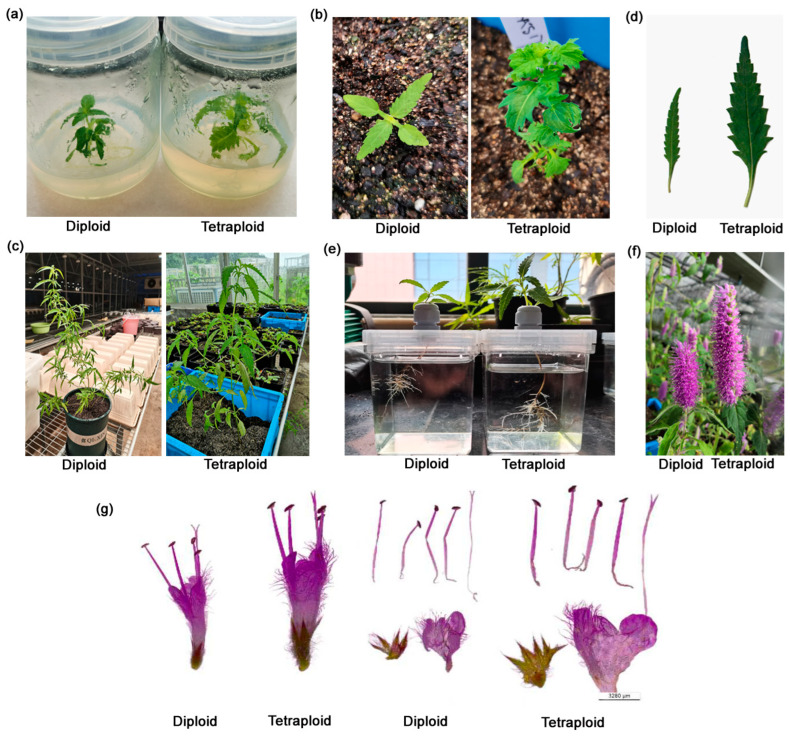
Phenotype characteristics and chromosome number of diploid and tetraploid *E. splendens*. (**a**) A comparison of phenotype characteristics of different ploidy-level *E. splendens* plants at tissue culture stage. (**b**) Different ploidy-level *E. splendens* plants were transplanted into soil. (**c**) Phenotype characteristics comparison of different ploidy-level *E. splendens* plants at adult stage. (**d**) Leaf morphology of two different ploidy-level *E. splendens* plants at adult stage. (**e**) Tetraploid plants produced more lateral roots. (**f**) Tetraploid inflorescence displayed increased length. (**g**) Anatomical diagram comparison of one floret between diploid and tetraploid.

**Figure 2 plants-13-03374-f002:**
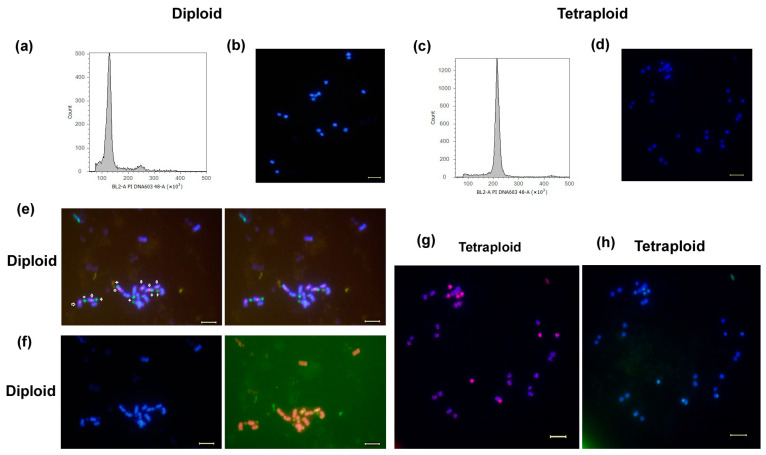
FCM and karyotyping identification of diploid and tetraploid *E. splendens*. (**a**) Ploidy level of diploid plants determined by FCM. (**b**) Ploidy level of diploid plants determined by karyotype. (**c**) Ploidy level of tetraploid plants as determined by FCM. (**d**) Ploidy level of diploid plants determined by karyotype. (**e**) Telomere repeat probe signal in diploid. (**f**) 5SrDNA 18SrDNA hybridization signal in diploid. Arrow represented 5SrDNA (red); “+” represented 18SrDNA (green). (**g**) Telomere repeat probe signal in tetraploid. (**h**) 5SrDNA hybridization signal in tetraploid. Bar = 5 µm.

**Figure 3 plants-13-03374-f003:**
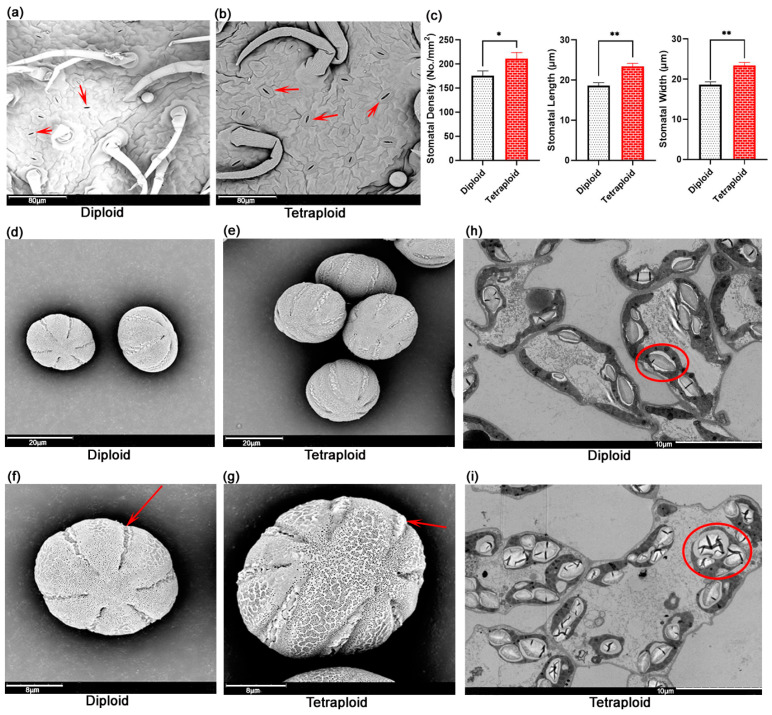
Microstructural and ultrastructural comparison between diploids and tetraploids. (**a**) Electron microscope structure of diploid *E. splendens* leaves. The red arrow points to the location of the stomata. (**b**) Electron microscope structure of tetraploid *E. splendens* leaves, the red arrow points to the location of the stomata. (**c**) Comparison of stomatal density, length, width between diploid and tetraploid *E. splendens*. Error bars indicate SE. Asterisks indicate significant differences between diploid and tetraploid *E. splendens* (*: *p* < 0.05; **: *p* < 0.01). (**d**) Electron microscope structure of diploid *E. splendens* pollen. (**e**) Electron microscope structure of tetraploid *E. splendens* pollen. (**f**) Electron microscope structure of a single diploid *E. splendens* pollen. The red arrow points to the location of the colporate. (**g**) Electron microscope structure of a single tetraploid *E. splendens* pollen. The red arrow points to the location of the colporate. (**h**) Chloroplast ultrastructure of diploid *E. splendens* leaves. The red circle represents a chloroplast. (**i**) Chloroplast ultrastructure of tetraploid *E. splendens* leaves. The red circle represents a chloroplast.

**Figure 4 plants-13-03374-f004:**
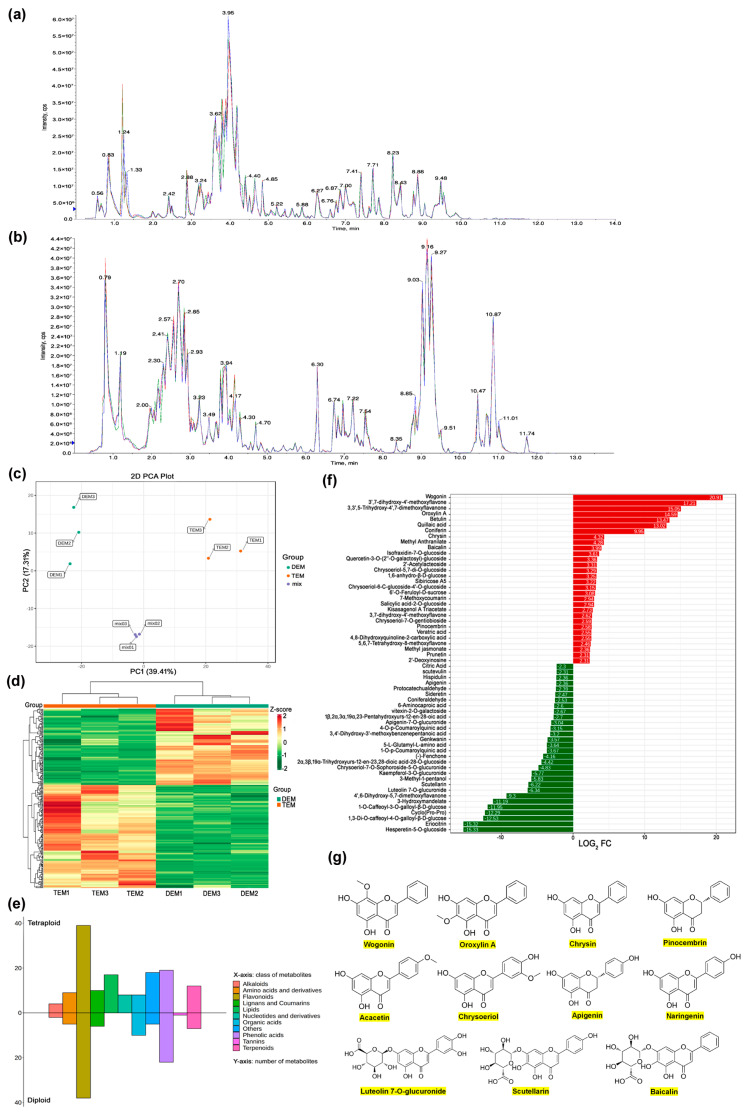
Widely targeted metabolomics comparison between diploids and tetraploids using UPLC-MS/MS. (**a**) Detection of total ion chromatogram (TIC) overlap map of QC samples in negative ion mode. (**b**) Detection of TIC overlap map of QC samples in positive ion mode. (**c**) PCA diagrams of diploid samples, tetraploid samples and QC samples. (**d**) A hierarchical clustering heat map of differentially accumulated metabolites (DAMs) between diploids and tetraploids. (**e**) Comparison plot of the classification of differential metabolites between diploids and tetraploids. (**f**) Bar chart of top 30 DAMs in diploids and tetraploids. Red represents DAMs in tetraploids, green represents DAMs in diploids, numbers represent value of log_2_ fold change. (**g**) Structure of main polyphenol metabolites detected in diploids and tetraploids.

**Figure 5 plants-13-03374-f005:**
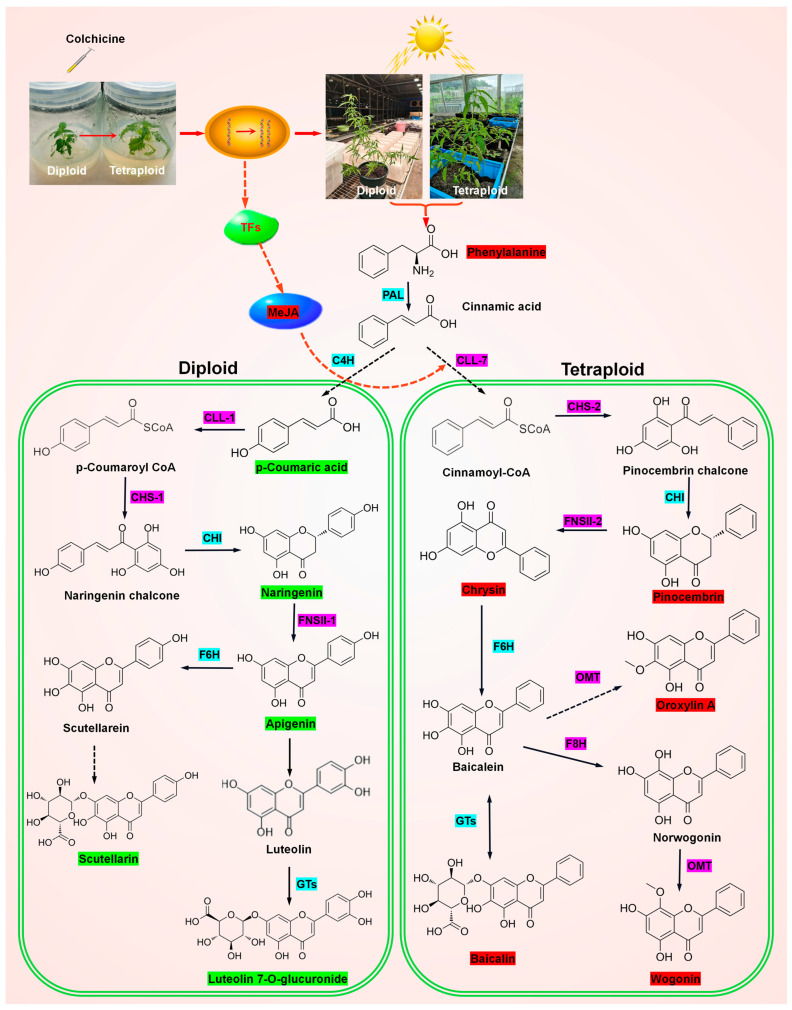
Schematic diagram of the mechanism by which ploidy affected polyphenol species in *E. splendens*. The red highlighted compound represents significantly accumulated metabolites in tetraploids. The green highlighted compound represents significantly accumulated metabolites in diploids. The unhighlighted compound represents no differences in metabolites between diploids and tetraploids. The magenta highlighted compound represents differential functional enzymes between diploids and tetraploids. The cyan highlighted compound represents no difference in enzymes between diploids and tetraploids.

**Table 1 plants-13-03374-t001:** Screening of the optimal axillary bud regeneration media for *Elsholtzia splendens*.

6-BA (mg/L)	NAA (mg/L)	Regeneration Rate (%)	Shoots Per Explant (No.)
0.1	0.01	84.59 ± 0.86 ^c^	3.25 ± 0.18 ^c^
	0.03	51.63 ± 1.38 ^f^	2.64 ± 0.06 ^d^
	0.06	48.52 ± 0.84 ^g^	2.81 ± 0.10 ^d^
	0.1	34.21 ± 2.55 ^h^	1.57 ± 0.08 ^f^
0.3	0.01	53.56 ± 0.83 ^ef^	3.23 ± 0.06 ^c^
	0.03	95.72 ± 1.23 ^a^	4.88 ± 0.13 ^a^
	0.06	89.56 ± 0.91 ^b^	4.16 ± 0.12 ^b^
	0.1	66.57 ± 0.55 ^d^	1.20 ± 0.15 ^g^
0.6	0.01	17.53 ± 1.27 ^j^	0.98 ± 0.89 ^g^
	0.03	25.26 ± 2.16 ^i^	1.96 ± 0.06 ^e^
	0.06	55.26 ± 1.09 ^e^	2.68 ± 0.11 ^d^
	0.1	47.53 ± 1.88 ^g^	3.48 ± 0.18 ^c^
1	0.01	0.00 ± 0.00 ^k^	0.00 ± 0.00 ^h^
	0.03	0.00 ± 0.00 ^k^	0.00 ± 0.00 ^h^
	0.06	0.00 ± 0.00 ^k^	0.00 ± 0.00 ^h^
	0.1	21.66 ± 2.15 ^k^	1.08 ± 0.95 ^g^

These data represent the mean ± SE of 10 replicates. The different letters in the same column data indicate significant differences based on *t*-test at the 0.05 probability level.

**Table 2 plants-13-03374-t002:** Screening of the optimal rooting medium for *E. splendens*.

MS	IBA (mg/L)	NAA (mg/L)	Roots Number Per Explant (No.)	Optimal Rooting Days (d)	Transplant Survival Rate (%)
MS	0.0	0.0	8.51 ± 0.56 ^k^	38.47 ± 0.58 ^e^	98.22
	0.0	0.05	12.24 ± 0.23 ^gh^	35.56 ± 1.06 ^f^	80.25
	0.0	0.1	16.35 ± 0.27 ^d^	34.65 ± 0.94 ^f^	60.76
MS	0.05	0.0	10.16 ± 0.81 ^ij^	30.87 ± 0.57 ^g^	78.50
	0.05	0.05	15.77 ± 0.34 ^d^	28.35 ± 0.76 ^h^	58.49
	0.05	0.1	18.61 ± 0.62 ^c^	22.84 ± 0.12 ^j^	56.04
MS	0.1	0.0	15.15 ± 0.14 ^e^	24.28 ± 0.65 ^i^	60.28
	0.1	0.05	21.91 ± 0.75 ^b^	20.56 ± 0.98 ^k^	46.20
	0.1	0.1	25.34 ± 0.15 ^a^	16.67 ± 1.15 ^l^	35.87
1/2 MS	0.0	0.0	9.25 ± 0.24 ^jk^	60.73 ± 1.02 ^a^	100.00
	0.0	0.05	11.46 ± 0.35 ^hi^	55.48 ± 0.85 ^b^	98.74
	0.0	0.1	13.08 ± 0.45 ^f^	50.26 ± 0.97 ^c^	86.35
1/2 MS	0.05	0.0	11.82 ± 0.15 ^h^	56.47 ± 0.62 ^b^	95.94
	0.05	0.05	12.64 ± 0.87 ^fgh^	50.72 ± 1.09 ^c^	60.38
	0.05	0.1	16.27 ± 0.65 ^d^	42.21 ± 0.46 ^d^	56.74
1/2 MS	0.1	0.0	12.85 ± 0.43 ^fg^	38.77 ± 1.05 ^e^	58.35
	0.1	0.05	15.91 ± 0.15 ^d^	30.34 ± 0.63 ^g^	55.47
	0.1	0.1	19.24 ± 0.61 ^c^	26.73 ± 1.07 ^h^	43.98

These data represent the mean ± SE of 10 replicates. The different letters in the same column data indicate significant differences based on T-test at the 0.05 probability level.

**Table 3 plants-13-03374-t003:** Screening of the optimal mutagenesis medium for *E. splendens*.

Treatments		Survival Rate of 50 Buds (%)	Number of Chimera and Diploid (No.)	Number of Tetraploid (No.)	Number of Tetraploids Surviving Transplantation (No.)
Colchicine Concentration (%)	Time (h)				
0.02	6	100.00	50	0	0
	12	100.00	50	0	0
	24	95.34	45	3	3
	48	82.63	36	5	5
0.05	6	100.00	43	7	7
	12	96.47	35	13	10
	24	86.35	33	10	9
	48	60.56	15	15	10
0.1	6	100.00	34	16	13
	12	95.56	18	30	27
	24	88.56	14	40	34
	48	45.63	5	17	10
0.2	6	60.12	8	22	17
	12	21.33	1	9	2
	24	5.62	0	3	0
	48	0.00	0	0	0

**Table 4 plants-13-03374-t004:** Effect of ploidy level on different phenotypic characteristics of *E. splendens*.

Characteristics	Diploid	Tetraploid	Significance
Plant height (cm)	60.88 ± 7.56	135.86 ± 13.56	**
Number of branches	8.53 ± 1.96	14.66 ± 2.14	*
Individual dry weight (g)	2.10 ± 1.19	26.79 ± 4.43	***
Fresh weight (g)	327 ± 19.567	1428 ± 29.51	****
Root length (g)	18.62 ± 7.25	41.64 ± 11.56	*
Shoot length (cm)	43.28 ± 5.37	93.66 ± 7.22	***
Shoot diameter (mm)	2.33 ± 0.18	5.89 ± 0.42	***
Leaf length (cm)	5.88 ± 0.48	8.85 ± 0.43	**
Leaf width (cm)	1.35 ± 0.07	2.53 ± 0.17	***
Leaf length/leaf width ratio	4.36 ± 0.26	3.50 ± 0.15	**
Specific leaf area (SLA) (cm^2^/g)	2.43 ± 0.13	0.59 ± 0.22	***
Specific leaf weight (SLW) (g/cm^2^)	0.41 ± 0.14	1.69 ± 0.21	***
Flower length (cm)	5.88 ± 0.48	8.85 ± 0.53	**
Flower width (cm)	1.35 ± 0.22	2.53 ± 0.37	**

These data represent the mean ± SE of 10 replicates. Asterisks indicate significant differences between diploid and tetraploid *E. splendens* (*: *p* < 0.05; **: *p* < 0.01; ***: *p* < 0.001; ****: *p* < 0.0001).

## Data Availability

Data are contained within the article and Appendix A.

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
