# Peer review of "In Vitro Inducted Tetraploid *Elsholtzia splendens* Nakai ex F. Maek. Alters Polyphenol Species and Synthesis"

_plants, 2024, doi:10.3390/plants13233374_

Round 1

Reviewer 1 Report

Comments and Suggestions for Authors

The manuscript plants-3321164 is comprehensive, presenting a lot of results that could successfully fill two research articles. Its great merit is the coherence of the research carried out and the attempts to answer sequentially emerging questions. The selection of appropriate conditions for polyploidisation is valuable; however, for me, the most important is the demonstration of differences in the expression of synthesis pathways of individual polyphenols of biological significance. These data can support research efforts introducing polyploidisation into the modern cultivation of pharmacologically valuable plants.

The work is well prepared and edited. Its strengths are the numerous photographs, diagrams and drawings.

My comments concern only minor errors, omissions and certain inaccuracies.

I would consider whether to use the word "polyphenols" instead of "flavonoids" when analysing changes in the synthesis level. As the data in Fig. 5 show, differences also apply to phenolic acids, including p-coumaric and cinnamic acid derivatives.

The authors wrote two very similar sentences: “Elsholtzia splendens, an annual herb, possesses a vast range of applications, extending beyond soil remediation to encompass medicine, food, spices, detergents, animal feed, and other domains” (lines 8–11) and “It is evident that E. splendens possesses a vast range of applications, extending beyond soil remediation to encompass medicine, food, spices, detergents, animal feed, and other domains” (lines 40–42). These sentences, however understandable, are mental shortcuts. In my opinion, it would be elegant to write that the plant is used in the food industry, including animal feed, detergent production and as a spice. I would consider whether to convert “medical” use into “pharmaceutical” use. What did the authors mean when they wrote about "other domains"?

Latin genus names, including the genus Elsholtzia, should be in italics. See lines 29, 72, 75 and 78.

There is no explanation of the abbreviations: 6-BA, IBA and NAA.

In Fig. 1, the order of the photos is reversed. First we have (a), then (b), then (d), then (c), and further photos. A similar array is used in Figs. 2 and 3, but there are appear to be intentional.

There is an error in the sentence: “Table S1 Comparison of all detected metabolite informations between diploids and tetraploids in Elsholtzia splendens” because a noun “information” does not a plural form.

Explanatory notes to Table S1 are missing. It is not fully clear what the symbols in some columns mean and how the data should be compared (there is the phrase “comparison between diploids and tetraploids” in the title). Some of these parameters are described in the text, but as the table, especially one that is included in the Supplementary Materials, is somewhat self-contained, I am missing a legend for it.

I would have arranged the layout of the charts within Figure 4 slightly differently. Fig. 4f could have been enlarged slightly by the authors, as at 200% magnification the values on the bars are still poorly visible, as are the names of the compounds on its left-hand side. Perhaps changing the font colour from grey to black would suffice? Figures 4a, b, c and d would also benefit greatly if they were larger.

Nov. 11, 2024

Author Response

Responses to reviewers  #1’ comments:

        Our gratitude is extended to Reviewer #1 for your valuable and constructive suggestions, which have proved to be of significant assistance in improving the manuscript. Your feedback is invaluable in helping us to enhance the quality of our manuscript. In response to the suggestions provided, we have prepared a point-by-point response as follows:

  1. I would consider whether to use the word "polyphenols" instead of "flavonoids" when analysing changes in the synthesis level. As the data in Fig. 5 show, differences also apply to phenolic acids, including p-coumaric and cinnamic acid derivatives.

Re: We are grateful for your comment. In this study, flavonoids and phenolic acids represent the two primary categories of metabolites that differentiate diploids from tetraploids. In light of your suggestion, we also recognize that polyphenols encompass these two main categories of differences and are a more precise term. Consequently, amendments have been implemented in the pertinent sections (title, abstract, etc.) and the alterations have been highlighted. Concurrently, Figure 5 has been modified and annotations have been incorporated to enhance the figure's interpretation (Please see Line 415-422 in revised manuscript).

  1. The authors wrote two very similar sentences: “Elsholtzia splendens, an annual herb, possesses a vast range of applications, extending beyond soil remediation to encompass medicine, food, spices, detergents, animal feed, and other domains” (lines 8–11) and “It is evident that splendenspossesses a vast range of applications, extending beyond soil remediation to encompass medicine, food, spices, detergents, animal feed, and other domains” (lines 40–42). These sentences, however understandable, are mental shortcuts. In my opinion, it would be elegant to write that the plant is used in the food industry, including animal feed, detergent production and as a spice. I would consider whether to convert “medical” use into “pharmaceutical” use. What did the authors mean when they wrote about "other domains"?

Re: Thanks for your comment. For this section, we have revised the text in line with the article's focus. The revised version highlights the medicinal value and removes superfluous information. For details, please see the INTRODUCTION section, highlighted in yellow.

  1. Latin genus names, including the genusElsholtzia, should be in italics. See lines 29, 72, 75 and 78.

Re: Thank you for pointing this out. We've put genus names in italics in the revised manuscript and highlighted (Please see Line 32, 71, 74, 78 and 399 in revised manuscript).

  1. There is no explanation of the abbreviations: 6-BA, IBA and NAA.

Re: Thank you. N-(Phenylmethyl)-9H-purin-6-amine (6-BA), Indole-3-butanoic acid (IBA), 1-Naphthylacetic acid (NAA). We have added the full name where they first appeared and highlighted (Please see Line 95, 98 and 105 in revised manuscript).

  1. In Fig. 1, the order of the photos is reversed. First we have (a), then (b), then (d), then (c), and further photos. A similar array is used in Figs. 2 and 3, but there are appear to be intentional.

Re: I am grateful for your meticulous examination of the issue. The images that are currently displayed have been ordered in accordance with the sequence in which they appear in the article. Nevertheless, the actual sequence of the images will result in a size that is not optimal for the width of the layout. Consequently, modifications have been implemented with the objective of enhancing the visual presentation and ensuring an aesthetically pleasing composition of the whole. Whether it would be possible to retain the existing image configuration?

  1. There is an error in the sentence: “Table S1 Comparison of all detected metabolite informations between diploids and tetraploids inElsholtzia splendens” because a noun “information” does not a plural form.

Re: Thanks for your friendly reminder. We've corrected it in the revised manuscript and supplementary table 1.

  1. Explanatory notes to Table S1 are missing. It is not fully clear what the symbols in some columns mean and how the data should be compared (there is the phrase “comparison between diploids and tetraploids” in the title). Some of these parameters are described in the text, but as the table, especially one that is included in the Supplementary Materials, is somewhat self-contained, I am missing a legend for it.

Re: Thanks for your comment. Explanatory notes to Table S1 have been added at the bottom of the table.

  1. I would have arranged the layout of the charts within Figure 4 slightly differently. Fig. 4f could have been enlarged slightly by the authors, as at 200% magnification the values on the bars are still poorly visible, as are the names of the compounds on its left-hand side. Perhaps changing the font colour from grey to black would suffice? Figures 4a, b, c and d would also benefit greatly if they were larger.

Re: Thank you. Figure 4 has been rearranged and enlarged in order to improve clarity.

Reviewer 2 Report

Comments and Suggestions for Authors

Dear authors,

Your work is very interesting and significant in both fundamental and practical aspects.

The work is very well organized and structured, the analyses and interpretations are well performed and vizualized.

I have some recommendations as follows:

1) Title and the first mention into the text: Please, add the author's name (Nakai ex F.Maek.) after the Latin name of the studied plant species

2) Line 29, 72, 75, 78, 387: The genus name should also be put in italic

3) Fig. 4 a, b, c, d, f are too small and hard to understand. Please, try to enlarge

4) Please, add some references for the methodology used in subsections 4.1, 4.2 and 4.4.

Author Response

Responses to reviewers  #2’ comments:

Reviewer #2:

    We are grateful for your meticulous examination of the manuscript. In accordance with the recommendations set forth, the manuscript has been revised in a systematic manner, with each recommendation addressed individually as follows:

  1. Title and the first mention into the text: Please, add the author's name (Nakai ex F.Maek.) after the Latin name of the studied plant species

Re: Thank you for your friendly comment. We have added the author's name after the Latin name in revised manuscript.

  1. Line 29, 72, 75, 78, 387: The genus name should also be put in italic

Re: Thank you very much. We've put genus names in italics and highlighted in the revised manuscript (Please see Line 32, 71, 74, 78 and 399 in revised manuscript).

  1. 4 a, b, c, d, f are too small and hard to understand. Please, try to enlarge

Re: Thanks for your suggestion. Figure 4 has been rearranged and enlarged to satisfy the reader's visibility.

  1. Please, add some references for the methodology used in subsections 4.1, 4.2 and 4.4.

Re: Thanks for your suggestion. Some references have been adopted in subsections 4.1, 4.2 and 4.4, and highlighted in revised manuscript.

Reviewer 3 Report

Comments and Suggestions for Authors

Dear Authors,

I have reviewed the manuscript and have the following observations:

The manuscript topic in Elsholtzia splendens provides new insights into the complex regulatory networks of ploidy-induced changes in metabolites.

The topic of the manuscript is good and forward-looking, and I recommend publication in the journal. 

Abstract: the aims and hypotheses are not formulated, so I suggest rewording this section.

Introduction chapter: the aims and hypotheses are not formulated, so I suggest rewording this section. 

Likewise, the conclusions do not state what the objectives were, what conclusions were drawn and why. No summary section, no implications for other researchers or for agriculture are described. I would suggest that this be written in several sections, especially in these chapters that I have highlighted. Because it is not coherent. 

Author Response

Responses to reviewers  #3’ comments:

Reviewer #3:

I have reviewed the manuscript and have the following observations: The manuscript topic in Elsholtzia splendens provides new insights into the complex regulatory networks of ploidy-induced changes in metabolites. The topic of the manuscript is good and forward-looking, and I recommend publication in the journal. Abstract: the aims and hypotheses are not formulated, so I suggest rewording this section. Introduction chapter: the aims and hypotheses are not formulated, so I suggest rewording this section. Likewise, the conclusions do not state what the objectives were, what conclusions were drawn and why. No summary section, no implications for other researchers or for agriculture are described. I would suggest that this be written in several sections, especially in these chapters that I have highlighted. Because it is not coherent.

Re: In response to the feedback provided by Reviewer #3, we would like to express our sincerest gratitude for the constructive suggestions you have offered. The feedback is of immense value in enabling us to enhance the quality of our manuscript. The text has been revised in accordance with the focus of the article in question. The revised version emphasises the medicinal value of the main metabolites and removes superfluous information, and described the significance of the experiment lies in obtaining new materials to enhance the value of Elsholtzia splendens in production. The manuscript has undergone a comprehensive revision, with particular attention paid to the abstract, introduction, and conclusion sections. The sections that have been revised are highlighted in yellow in the revised manuscript. We would be grateful for any further comments you may have on the revised manuscript. We would like to express our sincerest gratitude for your invaluable feedback again.